

# Dominant contribution of oxygenated organic aerosol to haze particles from real-time observation in Singapore during an Indonesian wildfire event in 2015

Sri Hapsari Budisulistiorini [1], Matthieu Riva [2,#], Michael Williams [2], Takuma Miyakawa [3], Jing Chen [1], Masayuki Itoh [4], Jason D. Surratt [2], Mikinori Kuwata[1,4,5]

[1]Earth Observatory of Singapore, Nanyang Technological University, Singapore 639798, Singapore
[2]Department of Environmental Sciences and Engineering, Gillings School of Global Public Health, The University of North Carolina at Chapel Hill, Chapel Hill, North Carolina 27599, United States of America
[3]Research and Development Center for Global Change, Japan Agency for Marine-Earth Science and Technology, Kanagawa 236-0001, Japan
[4]Center for Southeast Asian Studies, Kyoto University, Kyoto 6068501, Japan
[5]Campus for Research Excellence and Technological Enterprise (CREATE), Singapore 138602, Singapore
[#]Now at the Univ Lyon, Université Claude Bernard Lyon 1, CNRS, IRCELYON, F-69626, Villeurbanne, France

*Correspondence to*: S. H. Budisulistiorini (sri.hb@ntu.edu.sg); M. Kuwata (kuwata@ntu.edu.sg)

**Abstract.** Recurring transboundary haze from Indonesian wildfires in previous decades significantly elevated particulate matter (PM) concentrations in Southeast Asia. During that event on October 10 to 31, 2015, we conducted a real-time observation of non-refractory submicron PM (NR-PM$_1$) in Singapore using an Aerodyne aerosol mass spectrometer. Simultaneously, we characterized carbonaceous components and organic aerosol (OA) tracers from fine PM (PM$_{2.5}$) samples to support source apportionment of the online measurements. The real-time analysis demonstrated that OA accounted for approximately 80% of NR-PM$_1$ mass during the wildfire haze period. Source apportionment analysis applied to the OA mass spectra using multilinear-engine (ME-2) approach resulted in four factors: hydrocarbon-like OA (HOA), biomass burning OA (BBOA), peat burning OA (PBOA), and oxygenated OA (OOA). The OOA can be considered as a surrogate of both secondary organic aerosol (SOA) and oxidized primary organic aerosol (OPOA), while the other factors are considered as surrogates of primary organic aerosol (POA). The OOA accounted for approximately 50% of the total OA mass in NR-PM$_1$, while POA subtypes from wildfires (BBOA and PBOA) contributed to approximately 30% of the total OA mass. Our findings highlight the importance of atmospheric chemical processes, which likely include POA oxidation and SOA formation from oxidation of gaseous precursors, to the OOA concentration. As this research could not separately quantify the POA oxidation and SOA formation processes, further studies should attempt to investigate the contribution of gaseous precursors oxidation and POA aging to the OOA formation in wildfire plumes.

# 1 Introduction

Transboundary haze caused by wildfires has been a recurring issue in the South East Asia (SEA) for the past few decades, causing economic and health problems (Atwood et al., 2013; Engling et al., 2014; Heil and Goldammer, 2001; Nichol, 1997,



1998; Pavagadhi et al., 2013). Haze occurrence has been associated with anomalously low precipitation induced by the El Niño Southern Oscillation (ENSO) and the Indian Ocean Dipole (Field et al., 2009, 2016; Gaveau et al., 2015). Wildfires are especially pronounced in Indonesian tropical rainforests and peatlands due to land clearing activities, including extensive development of agricultural activities (Dennis et al., 2005; Murdiyarso et al., 2004; Siegert et al., 2001). During the 1997 El Niño period, the wildfires in Indonesia consumed both peat and surface vegetation and caused a severe transboundary haze (Heil and Goldammer, 2001).

A prolonged wildfire event occurred in Indonesia during September and October 2015, coinciding with an intense El Niño period (Field et al., 2016). The scale of this 2015 wildfire is thought to be the largest after the wildfire event in 1997, and was estimated to release 227 Tg of carbon into the atmosphere (Huijnen et al., 2016). These carbon emissions in 2015 were at least on an order of magnitude larger than those emitted by the 2013 wildfires in Indonesia (31 Tg of carbon; Gaveau et al., 2015), although lower than 800–2500 Tg of carbon released from the wildfire in 1997 (Page et al., 2002). By using atmospheric chemistry transport models, previous studies calculated the excess mortality rate associated with the 2015 wildfire haze between 11,000 to 100,000 individuals across Indonesia, Malaysia, and Singapore (Crippa et al., 2016; Koplitz et al., 2016). These model estimations already consider emission, atmospheric processing, and removal of gas and particulate matter (PM) emitted by the wildfires. However, atmospheric processing of aerosol particles in haze from Indonesian wildfires has scarcely been investigated. This lack of study inhibits a detailed treatment of atmospheric chemical processes in the models, including aerosol aging and secondary aerosol formation, in the models. Laboratory studies have shown that atmospheric processing of biomass burning smoke resulted in organic aerosol (OA) enhancement due to secondary organic aerosol (SOA) formation (Cubison et al., 2011; Ortega et al., 2013). Although SOA has been shown to significantly contribute to OA concentration in other parts of the world (e.g., Huang et al., 2014; Weber et al., 2007), contribution of SOA formation in haze particles from wildfires in the SEA has never been quantified.

Online aerosol measurement techniques, such as the Aerosol Mass Spectrometer (AMS) and the Aerosol Chemical Speciation Monitor (ACSM) developed by Aerodyne Inc. are useful for investigating atmospheric processing of aerosol particles, due to their high time resolution for chemical characterization of bulk aerosol composition (Jayne et al., 2000; Ng et al., 2011a). These techniques quantify the chemical characteristics of bulk OA allowing further multivariate factor analysis of the mass spectra (MS) matrix by positive matrix factorization (PMF) (Ulbrich et al., 2009; Zhang et al., 2011b) or multilinear engine (ME-2) solvers (Canonaco et al., 2013; Crippa et al., 2014). These multivariate factor analyses have been shown to be useful for identifying sources and atmospheric processes of OA, especially when combined with offline analytical techniques providing molecular-level chemical composition (e.g., Budisulistiorini et al., 2013; Zhang et al., 2005a).

In this study, we conducted in Singapore an atmospheric observation of haze particles induced by Indonesian wildfires using the time-of-flight ACSM (ToF-ACSM) from October 10 to 31, 2015. We characterized subtypes of ambient OA in the non-refractory PM$_1$ (NR-PM$_1$) using the Source Finder (SoFi) interface for the ME-2. In addition to the real-time measurements, we collected integrated filter samples over the course of the haze episode for elemental, organic and water-soluble organic



carbon (EC, OC andWSOC) analysis as well as molecular-level chemical characterization of OA constituents. Data from the ToF-ACSM and offline chemical analysis were combined for identifying and quantifying OA sources. We demonstrated in this study that Indonesian wildfires influenced the increasing NR-PM$_1$ and OA concentrations in Singapore, and atmospheric chemical processing likely contributed significantly to the total OA.

## 2 Methods

### 2.1 Measurement site

We conducted the 2015 haze measurement from October 10 to 31, 2015 on the fourth floor of North Spine building at Nanyang Technological University (NTU), Singapore (1.3483° N, 103.6831° E). Forested areas are located less than 1 km

north, west, and southwest of the sampling location. Jurong Island, where many chemical industries and petroleum refineries are located, is situated approximately 7 km south of the observation site. Traffic in the NTU campus is dominated by passenger vehicles and public transportation.

### 2.2 Real-time measurements by the ToF-ACSM

Ambient NR-PM$_1$ was measured continuously from October 10 to 31, 2015. Aerosols were drawn through 150 cm × 0.925

cm of outer diameter (0.622 cm of inner diameter) copper tubes from a PM$_{2.5}$ cyclone (URG, North Carolina). The residence time for PM$_{2.5}$ in the sampling inlet was ~1 sec at a sampling flow rate of 3 L min$^{-1}$. Aerosol-laden air was dried using an Aerodyne Dryer Module consisting of a multi-tube Nafion dryer (Perma Pure, New Jersey), which was operated in reflux mode. Under these operating conditions, the dried sampling flow for the ToF-ACSM was recycled as the sheath flow for the dryer to maintain the relative humidity (RH) of the sample below 20%. The aerodynamic lens mounted in the ToF-ACSM

allowed only PM$_1$ to penetrate into the vacuum chamber (Fröhlich et al., 2013, 2015). The ToF-ACSM data were obtained with a time resolution of 200 s using IgorDAQ v2.0.20 and the data were analyzed by Tofware version 2.5.6 written in Igor Pro (Wave Metrics Inc., Lake Oswego, Oregon). We calibrated sampling flow rate, ionization efficiency (IE) of nitrate, and relative IE (RIE) of ammonium and sulfate before and after the campaign. The calibration results before and after the campaign were consistent. A composition-dependent collection efficiency (Middlebrook et al., 2012) factor of 0.5 was

applied and confirmed by plotting total NR-PM$_1$ mass concentration measured by the ToF-ACSM against PM$_{2.5}$ mass concentration published on the website of the National Environment Agency (NEA) of Singapore (Fig. S1). The orthogonal linear regression analysis yielded a slope of 0.66 ± 0.01. We estimated PM$_1$ mass concentration by adding a concentration of elemental carbon (EC) from integrated sample measurements (details in Section 2.4.1) to the NR-PM$_1$ mass concentration. The orthogonal linear regression analysis of PM$_1$ versus PM$_{2.5}$ yielded a slope of 0.73 ± 0.05. A previous observation in

Singapore during haze episode showed that aerosol particles mainly compose of submicron particles (size range of ≤1.0 μm)



and only 1–4% of the fine aerosol mass are in the size range of 1.0–2.5 μm (Betha et al., 2014). Hence, the mass difference found in this study (0.2–0.3%) could not solely be attributed to particles with an aerodynamic diameter between 1 and 2.5 μm that were not measured in this study. A possible cause is collection efficiency of the ToF-ACSM system, which is similar to AMS system, decreases for particle size larger than 550 nm (Jayne et al., 2000). On the other hand, particles in the size range of 0.5–1.0 μm was a notable contributor (8–17%) to the total fine aerosol mass during haze episode (Betha et al., 2014). Hence, the low collection efficiency for size ranges of 0.5–1.0 μm might cause the ToF-ACSM measured less $PM_1$ mass than previous observation (~97% of $PM_{2.5}$ mass; Betha et al., 2014).

## 2.3 ME-2 analysis

The time series of OA mass spectra were deconvolved using an ME-2 solver, Source Finder (SoFi) tool version 6.2 written in Igor Pro (Canonaco et al., 2013). Details of the ME-2 analysis were described elsewhere (Canonaco et al., 2013; Crippa et al., 2014; Elser et al., 2016; Fröhlich et al., 2015) as well as in the SI (Section B). Briefly, a solution was first explored without any *a priori* information, meaning that no reference mass spectra were provided as constraints. Subsequently, mass spectra which originated from previous field and/or laboratory observations, were introduced to constrain one or more solution spectra. The strength of the constraints was adjusted by varying the degree of freedom ($\square$-value).

In this study, mass spectra from ambient measurements, such as hydrocarbon-like OA (HOA) and biomass burning OA (BBOA) (Ng et al., 2011c) and laboratory-generated peat burning OA (PBOA) (Budisulistiorini et al., 2017a), were employed to allow each factor to vary within a specific range. The factors solution variability ($\square$-value) ranged between 0 and 0.3 (with a 0.05 step) from the constraint/reference values. The optimum factors solution was selected by evaluating $Q/Q_{exp}$. $Q$ is the sum of the squared residuals weighted by their respective uncertainties (i.e., measurement and model), and $Q_{exp}$ is the degree of freedom of the model solution calculated based on the size of the data matrix and the number of factors (Canonaco et al., 2013; Ulbrich et al., 2009). The optimum factors solution was also examined by their correlations with chemical tracers and reference mass spectra. Analysis of the solutions is described in the supplementary information section (Tables S1-S4; Figs. S2-S4).

## 2.4 Integrated sample analyses

Ambient aerosols were sampled through a $PM_{2.5}$ cyclone (URG, North Carolina) at a total flow rate of approximately 16.7 L $min^{-1}$ connected to two 47 mm filter holders. The first filter holder was operated at a flow rate of 4 L $min^{-1}$ for ~24 h (starting around 08:00 Local Time (LT)) on October 14, 15 and 19-23, 2015. This filter holder hosted 47-mm Quartz fiber filters, which were pre-combusted at 900 °C for 3 h. Single-filter and dual-filter samples (consisting of front and back filters) were collected alternately over eight days. In total, only samples for eight days were collected because of a limited number of filters. These samples were analyzed for EC, organic carbon (OC), water-soluble organic carbon (WSOC) and inorganic ions, as described in section 2.4.1.



The second filter holder was operated at a flow rate of 4.2 L min$^{-1}$. The sampling duration was set for ~24 h when average hourly PM$_{2.5}$ reported in NEA website was less than 50 μg m$^{-3}$. The sampling duration was changed to ~12 h (08:00–19:45 for daytime and 20:00–07:45 for nighttime) when average hourly PM$_{2.5}$ was higher than 50 μg m$^{-3}$. In total, 26 filter samples were collected. Particles were collected on 47-mm Quartz fiber filters pre-baked at 250 °C for 16 h. The filters were baked at

different temperature settings, due to the availability of ovens. Filter samples from this second holder were analyzed for OA tracers, as described in section 2.4.2. All samples were covered by pre-baked aluminum foil and stored at -20 °C until they were analyzed. In addition to particle samples, field blanks were collected by putting filters in the holders for 5-10 min when the sampling pump was turned off. Field blanks were stored in the same manner to those of particle samples until analysis.

**2.4.1 OC, EC, WSOC and inorganic ions analyses**

EC and OC were quantified using a semi-continuous field type EC/OC analyzer (Sunset Laboratories Inc.), following the IMPROVE-A protocol (Miyakawa et al., 2015). The instrument was calibrated using a standard sucrose solution. The particulate OC was estimated by subtracting the back-filter OC from the front-filter OC, assuming that (1) particulate OC was collected by the front-filter, and (2) the gas-phase OC was collected on both the front- and back-filters. Analysis of WSOC was performed using a TOC analyzer (Model 800, Sievers, Colorado), following extraction of a portion (8 mm in

diameter) of filter samples by 10 mL of HPLC-grade water (Kanto Chemical Co. Inc., Tokyo, Japan) and using an orbital shaker for 21 h. The extracted sample solutions were filtered by syringe filters (PTFE; pore size 0.2 μm) prior to injection to the TOC analyzer. Possible interference of gaseous species on WSOC was also subtracted using the data from back-filters, as in the case of the OC analysis.

For the inorganic ions analysis, a portion of the filter samples (1 cm x 1 cm) was soaked in a solvent (4.75 mL MilliQ and

0.25 mL Ethanol). Ions were extracted by ultrasonication for 15 min., filtered by syringe filters (hydrophilic PVDF; pore size 0.45 μm), and were subsequently analyzed using ion chromatography (IC, Dionex ICS 1000, Thermo Fisher Scientific K.K.) (Miyakawa et al., 2015).

**2.4.2 OA tracers analysis by chromatography techniques**

The second set of samples was analyzed by gas and liquid chromatography techniques interfaced to mass spectrometry for

molecular-level analysis. Details of the chromatography techniques for OA tracers analysis have been reported in Budisulistiorini et al. (2017a). Briefly, the liquid chromatography system consists of liquid chromatography/diode array detector-electrospray ionization high-resolution quadrupole time-of-flight mass spectrometry (UPLC/DAD-ESI-HR-QTOFMS, hereafter called LC/MS; 6520 Series, Agilent, California) using a Waters ACQUITY UPLC HSS T3 column (2.1 × 100 mm, 1.8 μm particle size). The LC/MS samples were prepared using a quarter of the particle-laden filters. The samples

were extracted by adding 22 mL of methanol (LC/MS CHROMASOLV-grade, Sigma Aldrich, Missouri) and then sonication for 45 min at room temperature. To estimate recovery efficiency, ketopinic and camphor sulfonic acids were



spiked onto a quarter of the filters prior to extraction for estimation of recovery efficiency. The methanol extracts were gently dried by blowing an $N_2$ stream at room temperature. The dried extracts were reconstituted using 150 μL of methanol, and were subsequently analyzed by the LC-MS operated in the negative ionization mode. Details of the LC-MS operating conditions are described elsewhere (Zhang et al., 2011a). Table S5 provides a list of standard compounds used for

quantification of OA tracers.

The gas chromatography/mass spectrometry system consisted of gas chromatography/electron ionization-mass spectrometry (GC/EI-MS, hereafter called GC/MS; 5890 Hewlett Packard (HP) Series II Gas Chromatograph) interfaced to an HP 5971A Series Mass Selective Detector. An Econo-Cap™ EC-5 column (30 m × 0.25 mm × 0.25 μm) was utilized for the analysis. The GC/MS samples were extracted and dried in the same manner to the LC-MS samples. The dried extracts were

trimethylsilylated by adding 100 μL of N,O-bis(trimethylsilyl)trifluoroacetamide (BSTFA) + trimethylchlorosilane (TMCS) (99 : 1, v/v , Supelco, Sigma-Aldrich, Missouri) and 50 μL pyridine (Sigma-Aldrich, Missouri), and subsequently heated at 70 °C for 1 h. Within 24 h following trimethylsilylation, samples were analyzed by the GC/MS operating at 70 eV. Operating conditions and temperature program are described elsewhere (Surratt et al., 2010). Standards used for quantifying OA tracers by GC/MS are available in Table S6. Filter blanks were subtracted from the sample to account for artifacts that

were not removed during the filter pre-baking process.

**2.5 Elemental analysis**

We measured elemental compositions (carbon, hydrogen, nitrogen, and sulfur) of peat and vegetation samples from Riau and Central Kalimantan, Indonesia, using a 2400 CHNS Elemental Analyzer (Perkin Elmer). Samples were dried at 105°C for 16 hours and subsequently pulverized. The instrument was calibrated using 2.5-Bis(5-tert-butyl-benzoxazol-2-yl)thiophene

(BBOT) prior to the analysis. Table S7 lists the peat and vegetation samples that were analyzed for their elemental composition. In total, 11 samples were analyzed, including eight samples of peat and three samples of vegetation.

**2.6 Public domain data and analysis**

The NOAA Hysplit back-trajectory model (Rolph et al., 2017; Stein et al., 2015) was used for estimating the origins of air masses. The trajectories were calculated from the observation site at 200 and 500 m above ground level (a.g.l.) on 06:00 LT

of each day during the entire measurement period. In addition, trajectories were also estimated for four time periods (06:00, 12:00, 18:00, and 00:00 LT) of October 19 and 20 from the observation site at 500 m a.g.l. Fire hotspots in Indonesia during October 2015 were retrieved from near real-time (NRT) Moderate Resolution Imaging Spectroradiometer (MODIS) Thermal Anomalies/Fire Locations Collection 6 processed by the Land, Atmosphere Near real-time Capability for EOS (Earth Observing System) Fire Information for Resource Management System (LANCE-FIRMS, 2015). The air quality monitoring

data, including 1-h $PM_{2.5}$, 8-h CO and $NO_2$, from the website of the National Environment Agency (http://www.haze.gov.sg/) were also used in the analysis. Meteorological data, including wind speed, precipitation, relative



humidity (RH), and temperature, were obtained from the website of the Meteorological Service Singapore (http://www.weather.gov.sg/).

# 3 Results

## 3.1 Air mass back-trajectories

The back-trajectories of air masses during the haze episode in Singapore are shown in Figure S5. Air masses originated from south and southeast of Singapore, likely carried haze from wildfires in Sumatra and Kalimantan, Indonesia. Towards the end of the sampling period (October 29–31), the wind direction shifted, transporting air masses from northeast and east of Singapore. Air masses during this period might have been more dominated by local emissions as well as regional emissions from Malaysia and the South China Sea (e.g., ships emission and sea spray) (Betha et al., 2014). Therefore, we classified the measurements period into Period 1 (P1: for October 10–28), and Period 2 (P2: for October 29–31) in accordance with the wind directions, respectively. The highest concentration of haze particles was observed during October 19 and 20 (hereafter referred to as the peak event). Back-trajectories of air masses arriving at Singapore during this period show that the air mass was influenced by wildfires in Kalimantan and Sumatra, Indonesia (Fig. S6).

## 3.2 Time series of aerosol species and meteorological parameters

Figures 1a–1c show a time series of meteorological data. Wind speed and temperature measured at the sampling site during the P1 period were on average $1.9 \pm 0.4$ m s$^{-1}$ and $29.6 \pm 2.1$°C, respectively (Table 1). Wind speed and temperature during the P2 period were slightly lower than those of the P1 period, although the data of both periods agreed within one standard deviation. Average RH during the P1 period was $78.2 \pm 10.5\%$, and average precipitation during the corresponding period was $11.2 \pm 14.1$ mm per day, which occurred between October 24 and 28. This result suggests that precipitation and wet deposition were negligible for most of the P1 period. Both average RH (84.6%) and precipitation (39.0 mm day$^{-1}$) were higher during the P2 period. This enhanced precipitation could contribute to decreasing aerosol concentration during the P2 period, in addition to the termination of wildfires in Indonesia and change of wind direction as indicated by back-trajectories and fire hotspots in Figure S5.

Figure 1d shows a time series of mass concentration of aerosol species measured by the ToF-ACSM as well as PM$_{2.5}$ reported by NEA. The average NR-PM$_1$ mass concentrations during P1 and P2 periods were 53.6 μg m$^{-3}$ and 26.0 μg m$^{-3}$, respectively (Table 1). The highest concentration of NR-PM$_1$ was observed during the P1 period accounting for 312.6 μg m$^{-3}$ on October 19 around 22:00 LT. OA was the major contributor to the ambient submicron aerosol, accounting for ~80% of the NR-PM$_1$, as shown in Figures 1e-1g. During the P1 period, OA reached the maximum concentration of 264.6 μg m$^{-3}$ (October 19), at around the same time when NR-PM$_1$ reached the maximum concentration (312.6 μg m$^{-3}$). The minimum



concentration of OA during the P1 period was 5.6 μg m$^{-3}$. During the P2 period, the maximum and minimum OA concentrations were 34.6 and 5.3 μg m$^{-3}$, respectively. The small difference (0.3 μg m$^{-3}$) between the minimum OA concentrations of the P1 and P2 periods implies that the background concentration at the sampling site was approximately 5 μg m$^{-3}$. This value is within the range of previous estimations of organic matter concentration in ambient fine aerosol during

clear days in Singapore (Engling et al., 2014; See et al., 2006).

Sulfate (SO$_4^{2-}$) was the most abundant inorganic species during the entire campaign (Figs. 1e–1g). The maximum concentration of SO$_4^{2-}$ (26.7 μg m$^{-3}$) was observed around the same time as the maximum concentration of NR-PM$_1$. Elemental analysis of peat samples from Kalimantan yielded a trace of sulfur of 0.1–0.3% weight (Fig. S7), which could contribute to SO$_4^{2-}$ formation by oxidizing SO$_2$ during the atmospheric transport process. Moreover, a previous study showed

an increase of SO$_2$ levels during a fire episode in Kalimantan, Indonesia (Hayasaka et al., 2014). Thus, the enhancement of SO$_4^{2-}$ might be caused by the oxidation of SO$_2$ originating from the wildfires in Indonesia.

Ammonium (NH$_4^+$) was the second most abundant inorganic species, contributed 11–12% of the total NR-PM$_1$ across the entire campaign. The average concentration of NH$_4^+$ during the P1 period (2.9 μg m$^{-3}$) was approximately three times higher than that of the P2 period (1.0 μg m$^{-3}$). Estimation of the ratio of measured NH$_4^+$ to neutralized NH$_4^+$ (NH$_4^+_{meas}$/NH$_4^+_{neu}$)

yielded in ratios of 1.0 ± 0.1 for the P1 period and 0.7 ± 0.3 for the P2 period (Table S8 and Fig. S8; SI Section F). The NH$_4^+_{meas}$/NH$_4^+_{neu}$ ratio during the P1 period suggests that the cations (NH$_4^+$) were in excess for neutralization of the anions (i.e., SO$_4^{2-}$, NO$_3^-$, and Cl$^-$) (Zhang et al., 2007a), which was caused by elevated NH$_3$ emission from the peatland fires (Stockwell et al., 2016).The NH$_4^+_{meas}$/NH$_4^+_{neu}$ ratio does not indicate that aerosol acidity (pH) during the 2015 haze episode is neutral. The pH values were roughly estimated by a thermodynamic model (ISORROPIA-II; Fountoukis and Nenes, 2007)

to be on average 1.2 ± 0.6 during the entire campaign (Table S9). The high standard deviation indicates the high uncertainties in the estimation which could be caused by missing NH$_3$ and organic water in the model's input data (Budisulistiorini et al., 2017b; Weber et al., 2016).

Nitrate (NO$_3^-$) contributed 4.1% and 5.5% to the total NR-PM$_1$ during the P1 and P2 periods, respectively (Figs. 1f–1g). During the P1 period, the NO$_3^-$ concentration reached a maximum of ~14 μg m$^{-3}$ and decreased substantially to

approximately 4 μg m$^{-3}$ during the P2 period. Hence, NO$_x$ originating from wildfire (Hayasaka et al., 2014), combined with enhanced NH$_3$ concentration (Stockwell et al., 2016) during the P1 period, could increase of NH$_4$NO$_3$ concentration, and thus increasing the NO$_3^-$ concentration.

Chloride (Cl$^-$) was a minor component of NR-PM$_1$ during the haze event, accounting for less than 1% of aerosol concentration during P1 and P2 periods (Figs. 1f-1g). The maximum concentrations during P1 and P2 periods were 2.4 μg m$^{-3}$

(average 0.3 μg m$^{-3}$) and 0.9 μg m$^{-3}$ (average 0.2 μg m$^{-3}$), respectively. The difference in Cl$^-$ maximum concentrations indicates the contribution of Cl$^-$ to the increase of NR-PM$_1$ during the haze event, as previously observed in Singapore and Indonesia (See et al., 2006, 2007).





### 3.2 Diurnal profiles of aerosol species

The average diurnal profile of NR-PM$_1$ during the P1 period shows that the concentrations typically peaked around 14:00 and 22:00 LT (Fig. 2a). The nighttime increase was possibly driven by the peak event on October 19-20 when transport of highly concentrated haze plume from wildfires in Indonesia arrived in Singapore (Fig. S9a). During the P2 period, the

average diurnal profile of NR-PM$_1$ increases in the afternoon, suggesting a formation of secondary aerosol species such as SOA and inorganics (e.g., SO$_4^{2-}$, NO$_3^-$). During the P1 period, a slight increase of OA concentration in the afternoon and a large increase at nighttime as shown in Fig. 2b suggest that the local formation of OA was limited during the P1 period. Additionally, we found that the peak event influenced the nighttime increase of OA concentration during the P1 period (Fig. S9b), indicating that the night peak could be attributed to the arrival of highly concentrated haze plumes from wildfires in

Indonesia. An increase of OA concentration in the afternoon during the P2 period indicates that the local formation of SOA at the observation site was significant.

The SO$_4^{2-}$ concentration peaked around 14:00 LT in the afternoon (Fig. 2d), which resulted from the oxidation of SO$_2$ by hydroxyl radicals (Margitan, 1984). The diurnal profile of SO$_4^{2-}$ during the P2 period—when wildfires were less influential—suggests the existence of other sources of SO$_2$, including vehicles, ships, industries, and/or volcanoes

(Carmichael et al., 2002; Carn et al., 2017; Klimont et al., 2013; Seinfeld et al., 2004; Streets et al., 2000). Dimethyl sulfide (DMS) from the ocean could also contribute to SO$_4^{2-}$ formation in a coastal environment (Hoffmann et al., 2016; Ohizumi et al., 2016) such as Singapore. Diurnal profiles of NH$_4^+$ during the P1 and P2 periods increase in the afternoon (Fig. 2c). During the P1 period, the concentration of NH$_4^+$ slightly increased at nighttime. This small nighttime increase was likely caused by the peak event (Fig. S9c), associated with the arrival of highly concentrated haze plumes from Indonesian

wildfires. The diurnal profile of NO$_3^-$ during the P1 period (Fig. 2e) shows an increase at nighttime, which was not caused by the peak event (Fig. S9e). This nighttime increase of NO$_3^-$ might be resulted from the shift of the gas-aerosol phase equilibrium of NO$_3^-$ towards the particle phase. The concentration of Cl$^-$ increased in the morning and nighttime during the P1 and P2 periods (Fig. 2f). The Cl$^-$ concentration during the P1 period was slightly higher than that of the P2 period, which suggests a slight influence of wildfires.

### 3.3 Offline analysis data

Table 2 summarizes the ratios of organic matter (OM, which is OA from ToF-ACSM) to OC (OM:OC), OC:EC, and WSOC:OC from the current and previous ambient measurements (Agarwal et al., 2010; Aggarwal et al., 2013; Balasubramanian et al., 2003; Engling et al., 2014; Jayarathne et al., 2018; Kunwar and Kawamura, 2014; Park and Cho, 2011; See et al., 2006, 2007; Ye et al., 2017; Zhang et al., 2017). The OM:OC ratio of the current study (~1.8) is slightly

higher than 1.6 estimated from urban air samples influenced by traffic emissions (Zhang et al., 2017). The OC:EC ratio was higher (~4.8) than previous measurements (1.1–3.4) during clear days (Engling et al., 2014; Zhang et al., 2017); however it was within the range of measurements (2.8–6.0) during hazy days (Balasubramanian et al., 2003; See et al., 2006). The





WSOC:OC ratio during the 2015 haze event (~0.6) is higher compared to ~0.3 of the 2000 haze event (Balasubramanian et al., 2003). However, since the OM:OC ratio in 2015 was also higher, the ratios of WSOC:OM during the haze episodes in 2015 and 2000 are not significantly different (approximately 0.4 in 2015 and 0.2 in 2000).

Figure S10 shows concentrations of OA tracer compounds and OC measured by the offline analysis, and OA measured by the ToF-ACSM. The OA concentration was averaged over filter sampling time. Brown carbon (BrC) is a sum of light-absorbing constituents identified in Budisulistiorini et al. (2017a). These tracers are used in identifying OA sources resolved by ME-2 analysis (Table S4).

## 4 Discussion

### 4.1 Characterization of OA sources

The average OA mass spectra during the entire campaign (P1 and P2 periods) are shown in Figure 3. The mass spectra are generally similar with subtle differences in the intensity of some ions signals, such as ions at $m/z$ 44 and $m/z$ 43. Using ME-2 analysis, we further investigated the different sources of OA, based on the covariant of their mass spectra and time series. The ME-2 analysis resolved approximately 91% of the OA mass. The rest of the OA mass (~9%) was unresolved (i.e., unexplained residual and noise in Fig. S3). An increasing number of factors did not improve deconvolution of the OA mass, as shown by insignificant decreases of the unexplained residual and noise for the more than four factors (Fig. S3) and similarities between the oxygenated organic aerosol (OOA) factors (Fig. S2). The four OA factors from the 2015 haze data sets include both primary and secondary factors. The hydrocarbon-like organic aerosol (HOA) from fossil fuel combustion, biomass burning organic aerosol (BBOA) and peat burning organic aerosol (PBOA) from non-fossil burning processes are surrogates of POA. The OOA can be considered as a surrogate of both SOA and oxidized POA (OPOA) (Donahue et al., 2009; Jimenez et al., 2009; Zhang et al., 2005a, 2007b).

Figures 4a and 4b show the mass spectra and time series of the OA factors, respectively. The HOA factor, which is identified by a distinctive signal at $m/z$ 57, is attributed to emissions from fossil fuel combustion processes, such as diesel exhaust and lubricating oil (Zhang et al., 2005a, 2005b). Previous studies reported good correlations ($R^2$ >0.7) between HOA and CO, $NO_x$ and EC, known tracers of primary emission (Zhang et al., 2005a, 2005b). EC measured from the integrated samples was not correlated with HOA ($R^2 = 0$; Table S4). The limited number of integrated filter samples (n = 8) and the long sampling time (t ~23 h) might have inhibited capturing the temporal variation of EC. The 8-h average CO concentration retrieved from the NEA website was weakly correlated ($R^2$ ~0.2; Table S4) with the HOA factor. The average CO concentration during the P1 period (~1 mg m$^{-3}$) was higher than during the P2 period (~0.6 mg m$^{-3}$), indicating that wildfires in Indonesia were the dominant source of CO during the observation period. Relatively better correlations between CO and PBOA, as well as OOA ($R^2$ ~0.4), prove that CO mostly originated from the wildfires.



The BBOA factor can be attributed to vegetative burning. The MS of BBOA shows significant ion signals at $m/z$ 60 and 73 (mostly $C_2H_4O_2^+$ and $C_3H_5O_2^+$, respectively). These ions were associated with levoglucosan and other monosaccharide derivatives produced from thermal decomposition of cellulose during the combustion of wood and peat (Bates et al., 1991; Budisulistiorini et al., 2017a; Iinuma et al., 2007; Simoneit et al., 1999). Hence, levoglucosan can be used as a tracer for identifying the BBOA factor from general biomass burning (e.g., Alfarra et al., 2007; Lanz et al., 2008), and PBOA from peat burning (Budisulistiorini et al., 2017a; Iinuma et al., 2007). A good correlation ($R^2$ ~0.6; Table S4 and Fig. 4b) between BBOA and levoglucosan measured from integrated samples proves that BBOA can be associated with biomass burning. Additionally, we found that PBOA correlates well with levoglucosan ($R^2$ ~0.7, Table S4). The good correlations of BBOA and PBOA factors with levoglucosan suggest that BBOA and PBOA factors can be associated with vegetative and vegetative-detritus burning, respectively.

We identified the PBOA factor by a good correlation ($R^2$ =0.97; Table S3) with the mass spectrum of laboratory peat burning aerosol (Budisulistiorini et al., 2017a). As it was constrained with freshly emitted aerosol from laboratory peat burning, PBOA can be categorized as a primary aerosol although it might be slightly oxidized during atmospheric transport. Prominent signals at $m/z$ 43, 55, 57, 60, and 73 of the PBOA mass spectrum suggest a combined characteristic of BBOA and HOA factors. However, the temporal variation of PBOA was different from those of HOA and BBOA. We found a good correlation ($R^2$ ~0.6, Table S4 and Fig. 4b) between PBOA and the sum of BrC constituents characterized from laboratory peat burning aerosol and the ambient samples of the 2015 haze event (Budisulistiorini et al., 2017a). A weaker correlation between the BrC constituents with the BBOA ($R^2$ ~0.4) and HOA ($R^2$ ~0.5) factors indicates that BrC might be more closely associated with peat burning than with other sources.

Potassium ($K^+$) has also been used as a tracer for vegetative burning; however, it can also originate from sea salt, soil resuspension, and fertilizers (Andreae, 1983; Urban et al., 2012; Watson et al., 2001; Watson and Chow, 2001). To remove the contribution of sea salt, we estimated the non-sea-salt $K^+$ (nss-$K^+$) concentration by assuming that (1) the chemical composition of sea-salt particles is similar to that of seawater, and (2) the sodium ($Na^+$) particles in the samples is mainly from seawater spray (e.g., Balasubramanian et al., 2003). A weak correlation ($R^2$ ~0.2; Fig. S11) between nss-$K^+$ and levoglucosan indicates the influence of other sources of potassium (e.g., fertilizers and soil resuspension from gardening) that might exist at or around the observation site during the P1 period (Pachon et al., 2013). Since we did not have $K^+$ measurements during non-hazy days of the P2 period, we could not confirm the influence of the non-burning sources. Further linear regression analysis between nss-$K^+$ and OA factors yielded weak or no-correlation ($R^2$ = 0.1-0.3). The weak correlations between nss-$K^+$ and the OA factors could be due to (1) the inability of integrated samples to capture the temporal variation of potassium, and/or (2) the contribution of other sources that could not be identified in this study. Therefore, $K^+$ might not be a good tracer for identifying BBOA and PBOA factors at this observation site.

The OOA component was identified by the prominent ion signal at $m/z$ 44 ($CO_2^+$) representing oxygenated compounds (e.g., carboxylic acids group) (Zhang et al., 2005a). The OOA factor correlates well with ($R^2$ = 0.6; Table S4 and Fig. 4b) the sum of $SO_4^{2-}$ and nitrate (secondary aerosols) suggesting that the OOA might be formed through SOA formation processes. The



OOA factor is more pronounced during the P1 period than during the P2 period, suggesting that volatile organic compounds (VOCs) emitted from wildfires contributed to SOA formation. In fact, some SOA precursors such as benzene, toluene, phenol, and isoprene (Claeys et al., 2004; Yu et al., 2014) were identified from field measurements in Kalimantan, Indonesia during the 2015 peatland fire (Stockwell et al., 2016). Additionally, water-soluble organic gases (WSOGs), such as glyoxal,

methylglyoxal, and isoprene epoxidiol (IEPOX)), produced from oxidation of VOCs could partition to aerosol liquid water, cloud droplets, or fog water and form SOA (Blando and Turpin, 2000; Ervens et al., 2011; Sareen et al., 2017).

## 4.2 Diurnal variations of OA sources

Figure 5 shows the average diurnal variations of the OA factors during the P1 and P2 periods. During the P1 period (Figs. 5a and 5c), HOA concentration increased in the morning (07:00–09:00) and evening (18:00–23:00) implying that this HOA

factor can be associated with morning and evening traffic. A small decrease of the night peak when the peak event (October 19-20) was excluded from diurnal profile calculation (Fig. S12a) proves that the night peak of HOA during the P1 period was mainly influenced by night traffic. Diurnal variations of BBOA and PBOA show significant peaks at nighttime driven by the peak event during the P1 period, although no substantial variations were observed during daytime (Figs. S12b-S12c). The OOA concentration started to increase around 08:00 and peaked around 14:00 LT (Fig. 5a), indicating the contribution

of photooxidation processes. The elevated OOA concentration at nighttime was also generated by the peak event on October 19–20 (Fig. S12d).

During the P2 period (Figs. 5b and 5d), the HOA concentration increased in the evening, while no significant change was observed in the morning. The HOA concentration in the morning of the P2 period was almost half of its concentrations in the afternoon and evening resulting in an insignificant concentration profile in the morning. The HOA enhancement in the

evening can be associated with an evening traffic. An increase of the HOA concentration in the late afternoon (14:00–16:00) could be caused by heavy vehicle traffic from construction sites within the NTU campus, as well as regular traffic relating to school activities. It should be noted that the P2 period was shorter (October 29-31) than the P1 period. Hence, the diurnal profile of OA factors might not capture the actual variation of OA sources during non-hazy days. The BBOA concentration increased at noontime, although no significant local source of biomass burning was available during the P2 period. The

BBOA enhancement at noontime could be due to the transport of biomass burning plume event from Peninsula Malaysia that might occur during the short P2 period (Miettinen et al., 2017; Smith et al., 2017). The PBOA concentration did not vary significantly during the P2 period, suggesting that it might not be produced locally. The OOA concentration peaked around 14:00 LT and decreased in the evening, indicating that during the P2 period, OOA was likely formed through photooxidation.

## 4.3 Contribution of OA sources

The average concentration of OA sources to ambient fine aerosol during the 2015 haze episode in Singapore is illustrated in Figure 6a. The average concentrations of three OA components—BBOA (5.7 µg m$^{-3}$), PBOA (7.1 µg m$^{-3}$), and OOA (21.4



µg m$^{-3}$)—were notably higher during the P1 period than during the P2 period (BBOA = 1.9 µg m$^{-3}$, PBOA = 1.1 µg m$^{-3}$, and OOA = 5.8 µg m$^{-3}$). On the other hand, the HOA concentration remained consistent around 4 µg m$^{-3}$ during both periods. The changes in average concentration of OA factors during the P1 and P2 periods were not similar to the changes in contributions of OA factors to the total OA masses. As illustrated in Figure 6b, the contribution of OOA to the total OA mass

was higher during the P1 period (50.7%) than during the P2 period (39.9%). Similarly, the PBOA contribution was also higher during the P1 period (16.8%) than the P2 period (7.2%). The higher contribution of OOA and PBOA during the P1 period suggests that the factors were influenced by Indonesian wildfires. The BBOA contribution was relatively similar (~14%) during both periods, although the average BBOA concentration was three-time higher during the P1 period than the P2 period. The consistent contribution of BBOA during both periods might indicate an influence of other sources of BBOA

(e.g., wildfires in Peninsular Malaysia (Miettinen et al., 2017; Smith et al., 2017)) in addition to wildfires in Indonesia. Additionally, locally emitted cooking OA (COA), which has been shown to have similar mass spectra to BBOA (Lee et al., 2015; Liu et al., 2017), could also contribute to BBOA during the P2 period. This observation of the average concentrations and contributions of OA factors indicate that (1) fossil fuel combustion was likely the main source of HOA at most of the observation period, and (2) Indonesian wildfires contributed to the elevated BBOA, PBOA and OOA concentrations during

the P1 period.

## 4.4 Chemical characteristics of OOA

The oxidation degree of the total OA (and the OA factors) may provide information of the OOA formation processes. Figure 7a illustrates the relationship between the mass fraction of $m/z$ 44 ($CO_2^+$) to the total OA ($f_{44}$) versus $f_{43}$ (mass fraction of $m/z$ 43, mostly $C_2H_3O^+$, Fig. S13, to the total OA). This plot, known as the Triangle plot (Ng et al., 2011b), illustrates an

evolution of OA in the plume. Most of the OA during the entire campaign extends from the bottom to the middle-top of the triangle region (Fig. 7a). This OA distribution indicates the transition of OA, from less- to more-oxidized OA. The distribution of less- to more-oxidized OA is also supported by the plot of $f_{44}$ versus $f_{43}$ of the OA factors. BBOA, PBOA, and HOA reside near the bottom of the triangle region, representing the less-oxidized OA factor that exhibits low carbon oxidation states ($\overline{OS_C}$ = −0.98 to −1.42). OOA, which exhibit the highest carbon oxidation state ($\overline{OS_C}$ = 0.21), resides near the

top, indicating the more-oxidized OA factor.

The evolution of OA is also indicated by the diurnal profile of $f_{44}$ (Fig. S14a). $f_{44}$ started to increase in the morning and reached a maximum in the afternoon during both the P1 and P2 periods, suggesting an influence of oxidation processes on OA composition during the daytime. Figure S14b shows that during the P1 period, $f_{43}$ decreased in the afternoon and increased in the evening. The same figure shows that during the P2 period, $f_{43}$ peaked in the early morning and afternoon.

Figure 7b shows a relationship between $f_{44}$ and $f_{60}$ (mass fraction of $m/z$ 60, $C_2H_4O_2^+$ to the total OA), which can serve as an indicator of the aging processes of OA in plumes emitted by biomass burning (Cubison et al., 2011). Ion markers of biomass burning ($f_{60}$) during the P1 period were larger than a typical background level of $f_{60}$ of ~0.3% (Aiken et al., 2010), while those during the P2 period were very close to the background level (Fig. S14c). The slope of the P1 period (−3.3) indicates a



more pronounced progression towards a higher $f_{44}$ (more oxidized aerosol), while $f_{60}$ only slightly decreased. The changes in $f_{44}$ versus $f_{60}$ are possibly due to the atmospheric aging processes, such as levoglucosan decay (Cubison et al., 2011; Hoffmann et al., 2010), as well as SOA formation processes that enhance ion signals at $m/z$ 44. The OOA factor was likely influenced by biomass burning, as it exhibits a slightly higher $f_{60}$ than that of the typical background level, although it is

lower than that of the BBOA and PBOA factors.

Figure 8 illustrates the ratios of H:C and O:C as well as average $\overline{OS_C}$ (Canagaratna et al., 2015; Kroll et al., 2011) of the bulk OA measurements and the OA factors on a van Krevelen (VK) diagram (Heald et al., 2010). Linear regression analyses of H:C versus O:C for the OA factors resulted in the slope of –0.06. The slope of the OA factors is close to 0, suggesting that OOA formation in Singapore during the 2015 haze event could be driven by OA aging through hydroxyl group

oligomerization in dark chemistry (Gilardoni et al., 2016), and/or the addition of highly oxidized carboxylic and/or peroxide groups (Kumar et al., 2016). The $\overline{OS_C}$ of OOA (0.21) shown in Figure 8 is similar to that of low-volatility OOA (LV-OOA, 0.25) described in a previous study (Canagaratna et al., 2015), suggesting that OOA was highly oxidized. The $\overline{OS_C}$ of OOA also closely agreed with that of laboratory-generated toluene-derived SOA (0.10) estimated by Canagaratna et al. (2015). The close agreement between OOA and laboratory studies indicates that SOA formation could potentially contribute to the

OOA factor, considering that SOA precursors—for instance, toluene, benzene, xylene, and other VOCs—were emitted at notable levels (0.1–1 g kg$^{-1}$) from wildfires in Kalimantan, Indonesia (Stockwell et al., 2016).

Overall, the observation of the 2015 haze episode in Singapore indicated the dominant contribution of OOA to the total OA. OOA formation could be caused by (1) oxidation of POA, and (2) the formation of SOA through oxidation of VOCs emitted from wildfires as well as partitioning of WSOGs onto aerosol liquid water or cloud droplets during atmospheric transport.

This present study, however, could not separately quantify the contribution of POA oxidation and SOA formation to the total OOA. Therefore, further studies are needed to investigate the contribution of SOA formation and POA oxidation to the OOA formation in the wildfire haze particles.

## 5 Conclusions

We investigated contributions of inorganic and organic particles to the Indonesian wildfire haze through directly measuring

NR-PM$_1$ in Singapore from October 10 to 31, 2015, and demonstrated the dominant presence of OOA. The measurement period was separated into the P1 period, which was influenced by Indonesian wildfires, and the P2 period, which was less influenced by the fires. The average submicron particle (NR-PM$_1$) concentration was notably higher (~54 µg m$^{-3}$) during the P1 period than during the P2 period (~26 µg m$^{-3}$). Similar trends were observed for OA, SO$_4^{2-}$, NH$_4^+$, and NO$_3^-$ characterized from the NR-PM$_1$. OA was the most abundant species measured during the entire campaign, averaging 42 µg m$^{-3}$ and 12 µg

m$^{-3}$ during the P1 and P2 periods, respectively. The SO$_4^{2-}$ concentration was higher during the P1 period (6 µg m$^{-3}$) than that of the P2 period (3 µg m$^{-3}$). The higher concentrations of OA and SO$_4^{2-}$ demonstrate the influence of wildfire emission.



Characterization of OA sources by ME-2 analysis yielded four components, i.e., HOA, BBOA, PBOA, and OOA, which can be associated with vehicles/traffics, wood/biomass burning, wildfires, atmospheric processing, respectively. OOA was the most abundant (~51% of OA), followed by PBOA (~17%), BBOA (~14%), and HOA (~10%). PBOA was associated with peat burning BrC constituents ($R^2$ ~0.6), while other OA factors showed less correlation. BBOA was characterized by correlating ($R^2$ ~0.6) with levoglucosan, which is a known wood/biomass burning tracer. An association between levoglucosan and OOA ($R^2$ ~0.6) during the P1 period and a notable decrease of OOA concentration in the P2 period, indicates the contribution of Indonesian wildfire emissions to OOA formation.

The relationships between $f_{44}$ versus $f_{60}$, as well as H:C versus O:C ratios of the OA factors, suggest that OOA formation in Singapore during the 2015 haze event could be driven by (1) POA aging, and (2) SOA formation from the oxidation of gas precursors and/or the aqueous phase chemistry of WSOGs during atmospheric transport. The significant contribution of OOA to the total OA warrants further study to investigate the contribution of oxidation of gas-phase precursors and aging of primary aerosols emitted from Indonesian wildfires.

### Acknowledgements

We acknowledge Dr. H. Gunawan for supporting our research in Indonesia. We thank G. B. Lebron and W. -C. Lee for assisting in the particle sampling and data collection, and K. Niezgoda and S. R. He for the meteorological data. We gratefully acknowledge the NOAA Air Resources Laboratory (ARL) for the provision of the HYSPLIT transport and dispersion model and/or READY website (http://www.ready.noaa.gov) used in this publication. We acknowledge the use of fire data from LANCE FIRMS operated by the NASA/GSFC/Earth Science Data and Information System (ESDIS) with funding provided by NASA/HQ. The research is funded by the National Research Foundation Singapore (NRF) under its Singapore National Research Fellowship scheme (National Research Fellow Award, NRF2012NRF-NRFF001-031), NRF Campus for Research Excellence and Technological Enterprise (CREATE) program, the Earth Observatory of Singapore (EOS), and Nanyang Technological University.

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

**Table 1: Average concentrations with one standard deviation (SD) of chemical species measured during the October 2015 haze episode in Singapore using ToF-ACSM (online) and integrated filter samples (offline) measurements. Wind speed and**
10  **precipitation data were retrieved from the website of Meteorological Service Singapore, and other meteorological parameters were from measurement at sampling location.**

|  |  | Overall |  | P1 |  | P2 |  |
|---|---|---|---|---|---|---|---|
| Parameters | Unit | Average | SD | Average | SD | Average | SD |
| *Online measurements* |  |  |  |  |  |  |  |
| NR-PM$_1$ | µg m$^{-3}$ | 49.8 | 27.7 | 53.6 | 26.9 | 26.0 | 0.4 |
| OA$^\#$ | µg m$^{-3}$ | 39.2 | 23.6 | 42.3 | 23.2 | 20.9 | 11.6 |
| HOA | µg m$^{-3}$ | 4.1 | 3.3 | 4.1 | 3.5 | 4.2 | 1.9 |
| BBOA | µg m$^{-3}$ | 5.3 | 2.8 | 5.7 | 2.7 | 2.0 | 1.2 |
| PBOA | µg m$^{-3}$ | 6.5 | 6.5 | 7.1 | 6.6 | 1.0 | 0.8 |
| OOA | µg m$^{-3}$ | 19.7 | 10.7 | 21.4 | 10.1 | 5.8 | 3.0 |
| NH$_4^+$ | µg m$^{-3}$ | 2.6 | 1.5 | 2.9 | 1.4 | 1.0 | 1.0 |
| SO$_4^{2-}$ | µg m$^{-3}$ | 5.6 | 3.0 | 6.0 | 2.7 | 2.5 | 2.4 |
| NO$_3^-$ | µg m$^{-3}$ | 2.1 | 1.4 | 2.2 | 1.4 | 1.4 | 0.7 |
| Cl$^-$ | µg m$^{-3}$ | 0.3 | 0.3 | 0.3 | 0.3 | 0.2 | 0.2 |
| *Offline measurements** |  |  |  |  |  |  |  |
| OC | µg m$^{-3}$ | 27.9 | 18.1 | 27.9 | 18.1 | n.a. | n.a. |
| EC | µg m$^{-3}$ | 6.0 | 0.7 | 6.0 | 0.7 | n.a. | n.a. |
| WSOC | µg m$^{-3}$ | 17.3 | 10.5 | 17.3 | 10.5 | n.a. | n.a. |
| Na$^+$ | µg m$^{-3}$ | 0.2 | 0.0 | 0.2 | 0.0 | n.a. | n.a. |
| K$^+$ | µg m$^{-3}$ | 0.5 | 0.1 | 0.5 | 0.1 | n.a. | n.a. |
| Mg$^+$ | µg m$^{-3}$ | 0.0 | 0.0 | 0.0 | 0.0 | n.a. | n.a. |
| Ca$^{2+}$ | µg m$^{-3}$ | 0.5 | 0.1 | 0.5 | 0.1 | n.a. | n.a. |
| *Meteorological conditions* |  |  |  |  |  |  |  |



| Parameters | Unit | Overall Average | SD | P1 Average | SD | P2 Average | SD |
|---|---|---|---|---|---|---|---|
| Wind speed | m s$^{-1}$ | 1.8 | 0.4 | 1.9 | 0.4 | 1.5 | 0.2 |
| Precipitation | mm | 10.3 | 7.9 | 11.2 | 14.1 | 39.0 | 6.9 |
| RH | % | 79.2 | 10.6 | 78.2 | 10.5 | 84.6 | 9.0 |
| Temperature | °C | 29.4 | 2.2 | 29.6 | 2.1 | 28.2 | 2.2 |

[#]*About 95% of OA matrix was resolved into four components by ME-2 analysis. *These measurements were conducted only during P1 period. Hence, the overall values were taken from P1 values. n.a. is not available.*

**Table 2: Ratios of OM:OC, OC:EC, and WSOC:OC of the atmospheric aerosols from urban and rural/background in Singapore and other Asian sites.**

| Type | Site | Period | OM:OC | OC:EC | WSOC:OC | Ref. |
|---|---|---|---|---|---|---|
| Urban | NTU[#] campus | Oct 2015 (haze) | 1.77 | 4.75 | 0.64 | This study |
| Singapore | Roadside | Nov 2015-Feb 2016 (clear) | 1.60 | 1.13 | 0.26 | a |
| | NUS[^] campus | Nov 2015-Feb 2016 (clear) | 1.60 | 3.36 | 0.69 | a |
| | NUS campus | Mar 2001-Mar 2002 (clear) | | 1.94 | | b |
| | NUS campus | Mar 2001-Mar 2002 (haze) | | 2.79 | | b |
| | NUS campus | Apr-May 2000 (haze) | 1.38 | 6.03 | 0.30 | c |
| Urban others | Changzou, China | Jul-Aug 2016 | | 5.00 | 0.58 | d |
| | Gwangju, South Korea | Jun-Aug 2008 | | 3.10 | 0.55 | e |
| | Mumbai, India | Jun 2006 | | 3.00 | 0.38 | f |
| | Sapporo, Japan | Aug 2005 | | 7.89 | 0.54 | g |
| | Pekanbaru, Indonesia | Mar 2005 (haze) | | 1.43 | | h |
| Rural/ background | St. John Island | Oct 2006 (haze, TSP[*]) | | 15.50 | | i |
| Singapore | St. John Island | Dec 2006-Jan | | 2.60 | | i |



| Type | Site | Period | OM:OC | OC:EC | WSOC:OC | Ref. |
|------|------|--------|-------|-------|---------|------|
| | | 2007 (clear, TSP) | | | | |
| Rural/ background others | Central Kalimantan, Indonesia | Nov 2015 (haze) | 1.26 | 67.00 | | j |
| | Okinawa Island, Japan | Jun-Aug 2010 | 2.01 | 9.42 | 0.29 | k |
| | Dumai, Indonesia | Mar 2005 (haze) | | 2.42 | | h |

Note: #NTU is Nanyang Technological University, ^NUS is National University of Singapore, *TSP is Total Suspended Particle. References: (a) Zhang et al. (2017); (b) See et al. (2006); (c) Balasubramanian et al. (2003); (d) Ye et al. (2017); (e) Park and Cho (2011); (f) Aggarwal et al. (2013); (g) Agarwal et al. (2010); (h) See et al. (2007); (i) Engling et al. (2014); (j) Jayarathne et al. (2017); (k) Kunwar and Kawamura (2014).

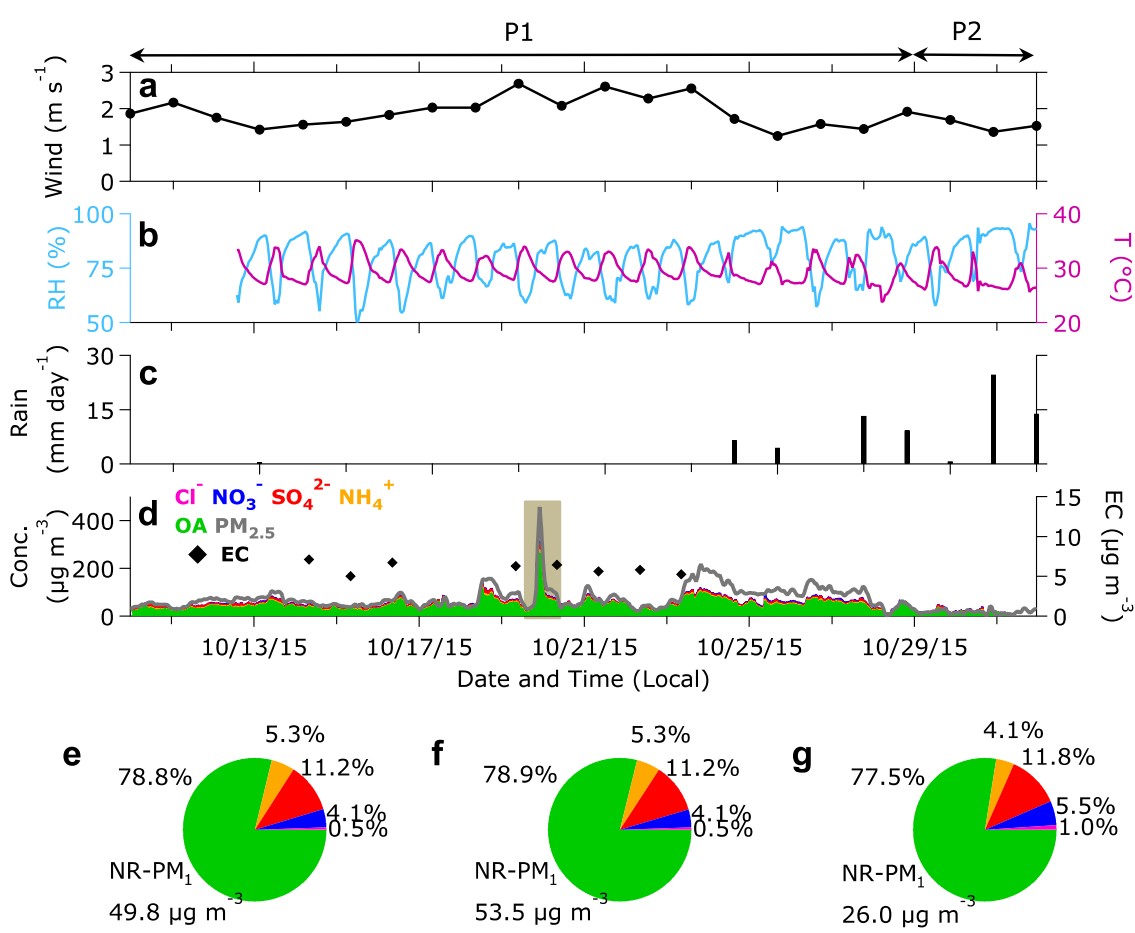



**Figure 1:** Time profiles of the meteorological parameters and aerosol concentrations. Daily (a) wind speed and (c) precipitation data retrieved from the Meteorological Service Singapore website, hourly (b) relative humidity (RH) and (c) temperature measured at NTU campus from October 10 to 31, 2015. (d) Time profiles of organic aerosol (OA), ammonium ($NH_4^+$), sulfate ($SO_4^{2-}$), nitrate ($NO_3^-$), and chloride ($Cl^-$), measured by the ToF-ACSM and total $PM_{2.5}$ mass published on the National Environment Agency website. Pie charts of ToF-ACSM species contribution to the total NR-$PM_1$ mass for (e) the overall, (f) P1 (October 10-28, 2015), and (g) P2 (October 29-31, 2015) periods are color coded similarly to the time profiles on plot (d). Shaded area on plot (d) is referred to as the peak event.

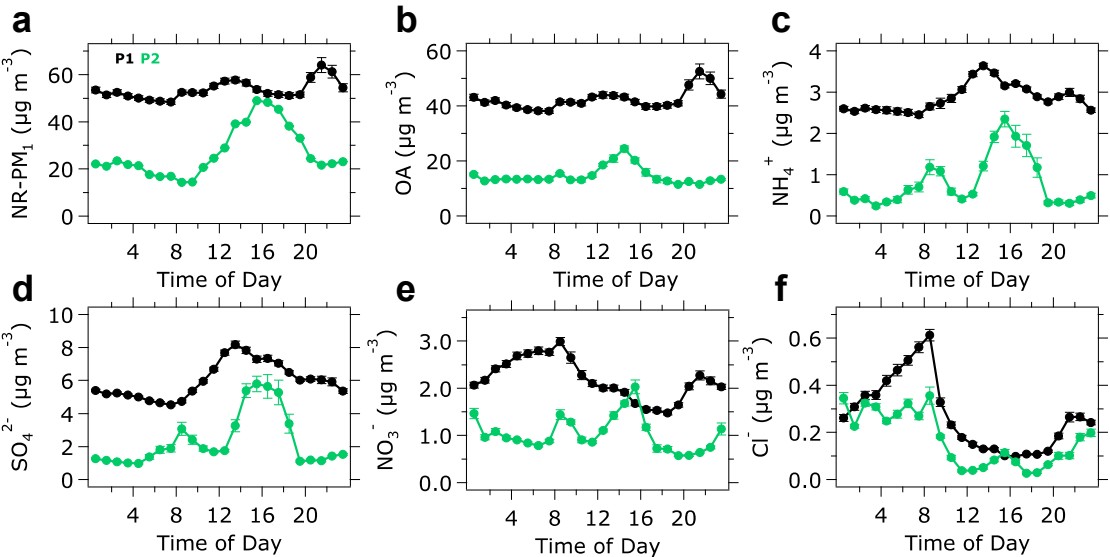

**Figure 2:** Average diurnal profiles of NR-$PM_1$ chemical species during the 2015 haze episode in Singapore. Vertical bars show one standard deviation.

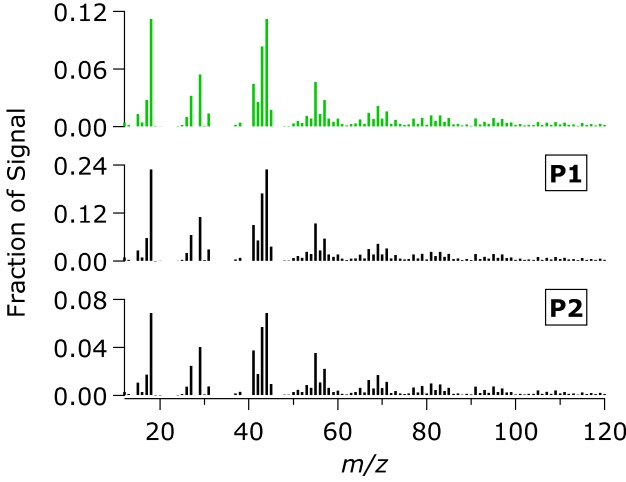

**Figure 3:** Average mass spectra of OA during the entire sampling period (top) and the P1 and P2 periods.



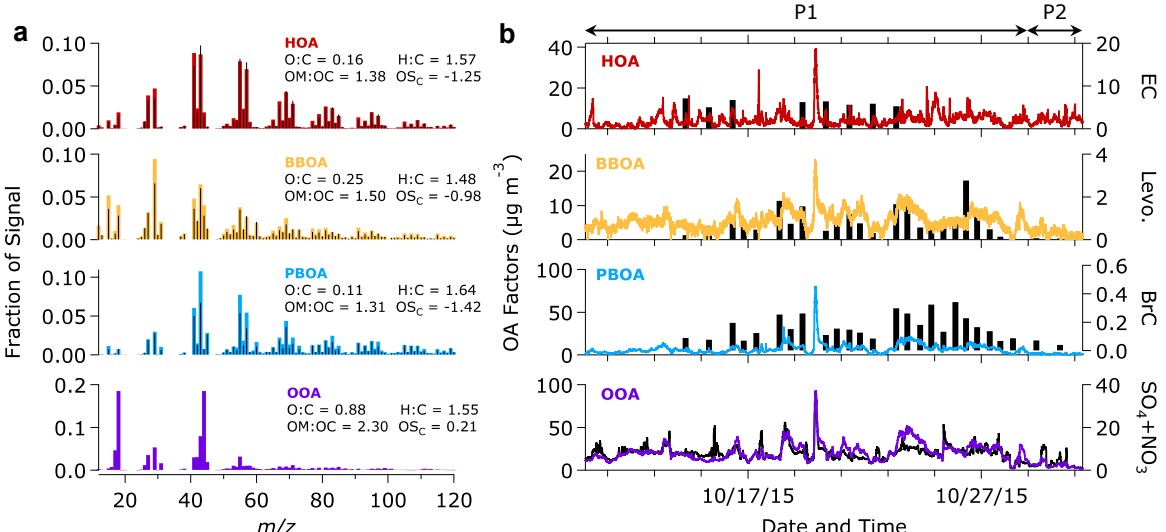

**Figure 4: (a) Mass spectra of OA components resolved by ME-2 analysis. Black bars represent the reference mass spectra of HOA and BBOA (Ng et al., 2011) and peat burning aerosol (Budisulistiorini et al., 2017) used as the constraints in ME-2 analysis. The elemental ratios were estimated from signals at mass-to-charge ratio (*m/z*) 44 ($f_{44}$) and 43 ($f_{43}$) using the improved-ambient**
5 **elemental analysis method (Canagaratna et al., 2015). (b) Time series of OA factors observed in Singapore during the 2015 haze episode. Time series of EC, levoglucosan, sum of brown carbon (BrC) constituents, and sum of $SO_4^{2-}$ and $NO_3^-$ in unit of ug m$^{-3}$ are shown against the time series of HOA, BBOA, PBOA, and OOA, respectively.**

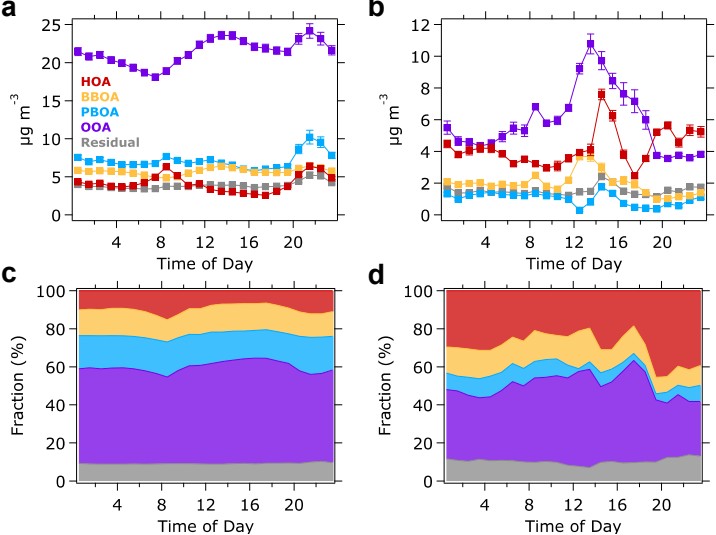

**Figure 5: Average diurnal profiles of OA sources in unit of ug m$^{-3}$ (top panels) for (a) the P1 and (b) P2 periods. Bottom panels**
10 **show similar diurnal profile but in fraction contribution to the total OA mass for (c) the P1 and (d) P2 periods.**





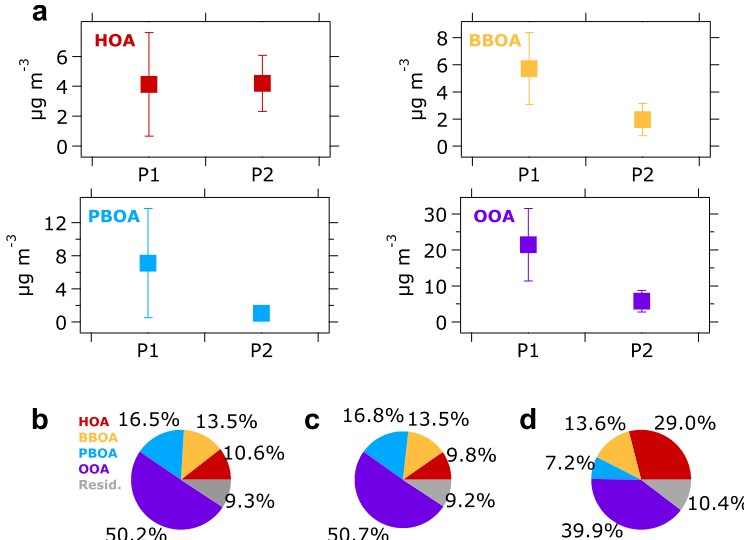

**Figure 6: (a) Average concentrations of OA factors during the P1 and P2 periods. Vertical lines represent one standard deviation. Contribution of each OA factor to the total OA mass during (b) the overall, (c) P1, and (d) P2 periods are illustrated in pie charts. Residual is the portion of time series OA mass spectra that could not be resolved by ME-2 solver.**

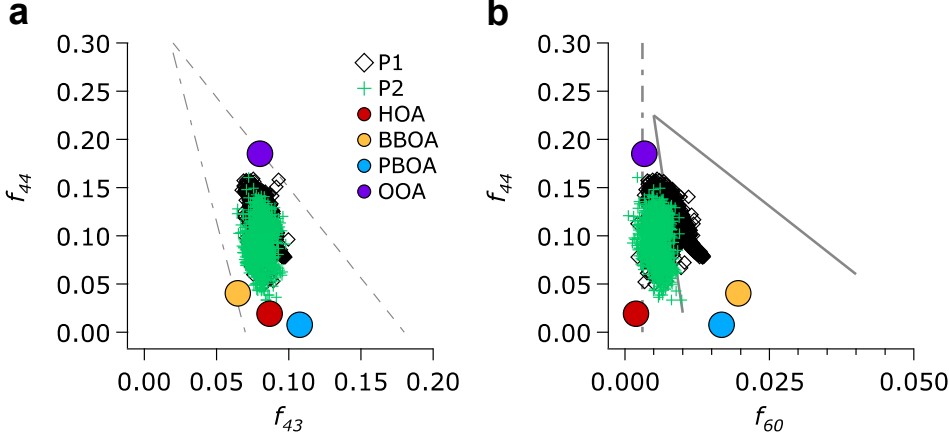

**Figure 7: (a) Scatterplot of $f_{44}$ (mass fraction of $m/z$ 44 to the total OA) as a function of $f_{43}$ (mass fraction of $m/z$ 43) of the OA and the factors resolved by ME-2 analysis. The grey dashed lines depicts the Triangle Plot (Ng et al., 2010) which is a guidance to show aging of the total OA (classified into the P1 and P2 periods) and the OA factors. (b) Scatterplot of $f_{44}$ as a function of $f_{60}$ of the total OA and the OA factors plotted over the conceptual space for biomass burning aerosol aging (solid grey lines) and the nominal background/urban value of $f_{60}$ at 0.3% (grey dashed and dot line) (Cubison et al., 2011).**





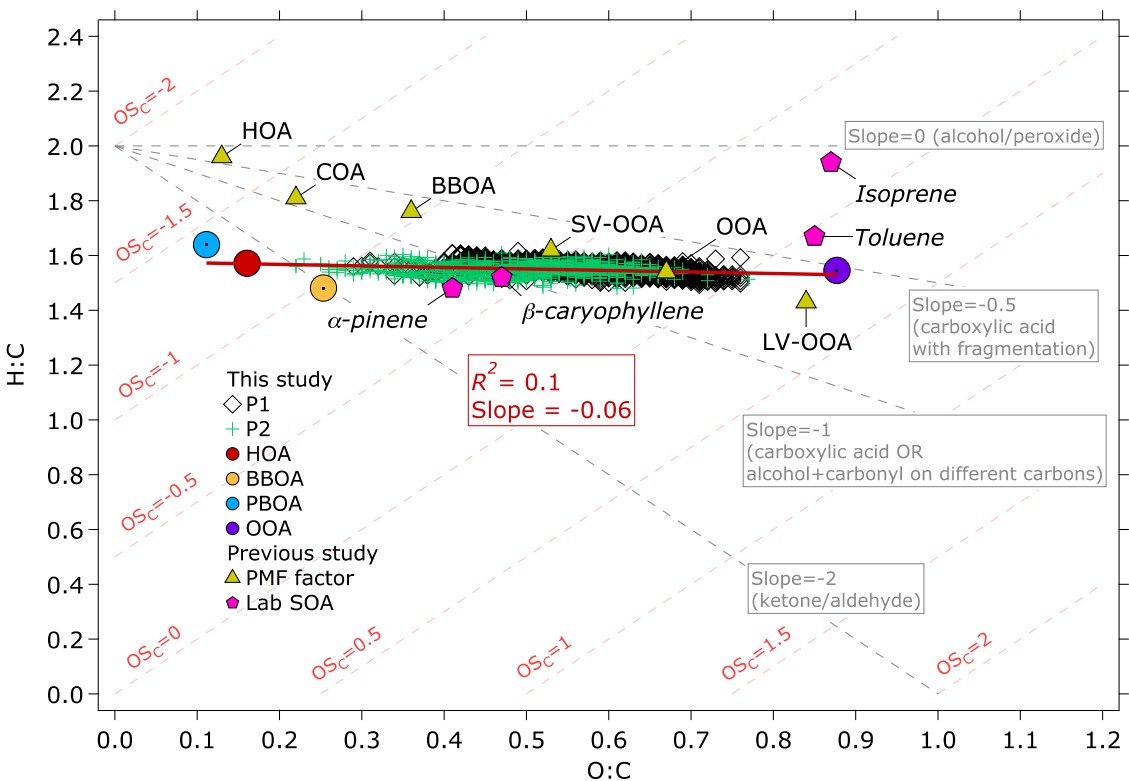

**Figure 8: Van Krevelen diagram of the OA measured during the 2015 haze episode and the OA factors resolved by ME-2 analysis. The H:C to O:C ratios are colored based on the P1 and P2 periods. The H:C to O:C ratios of the OA factors resolved by ME-2 analysis of this study are plotted as colored circles. Linear regression analysis of H:C vs. O:C from the OA factors yielded $R^2$ = 0.1 and slope = −0.06 as plotted on the figure. Comparison with HOA, BBOA, cooking OA (COA), OOA, semi-volatile OOA (SV-OOA), and low-volatility OOA (LV-OOA) (triangle markers) and laboratory-generated SOA (hexagonal markers) of previous studies (Canagaratna et al., 2015 and references therein) show that the OOA factor of this study exhibits close characteristics to LV-OOA and toluene-derived SOA, indicating that its formation could be attributed to atmospheric processing.**