# Peer review of "Dominant contribution of oxygenated organic aerosol to haze particles from real-time observation in Singapore during an Indonesian wildfire event in 2015"

_Atmospheric Chemistry and Physics, 2018_

## Referee Comment (RC1) · Anonymous Referee #1 · 19 Jun 2018

Review of "Dominant contribution of oxygenated organic aerosol to haze particles from real-time observation in Singapore during an Indonesian wildfire event in 2015" by Budisulistiorini et al.

The authors presented measurement results of particulate matter (PM) in Singapore affected by an Indonesian wildfire event in 2015. A suite of on-line and off-line measurement techniques were used to quantify chemical composition of PM. The major data from on-line ToF-ACSM measurements showed that oxygenated organic aerosol (OOA) contributed ~50% to total OA in non-refractory PM1 (NR-PM1), while biomass

burning and peat burning OA (BBOA and PBOA) contributed 30%, with the remaining 20% being hydrocarbon-like OA (HOA) or residual in PMF/ME-2 analysis. By combining on-line and off-line data, the authors concluded that both oxidation of gaseous precursors and aging of primary OA (POA) contributed to secondary OA (SOA) formation in the wildfire episode. For the POA aging, I incline to agree with the authors that it might occur, but I did not see direct evidence from the data (see Major comment #1 below). I would suggest the authors to either provide more evidence to support this statement or tune it down as a mere speculation as I see it.

The measurements and data analyses are rigorous and the manuscript is quite well written. The content is surely within the scope of ACP and is to the interest of readers of ACP. I therefore suggest minor revision with a few comments as below.

Major:

1. The assertion that SOA was contributed by POA (BBOA and PBOA) aging was only supported by an observation that the OOA factor has slightly higher f60 than normal background levels (page 14, line 3-5). While the OOA factor was surely influenced by BB or PB, whether the influence was merely factor mixing by PMF/ME-2 or truly due to aging processes is not clear. Set aside the fact that statistical analysis like PMF/ME-2 might suffer from factor mixing problems, correlation between OOA and BBOA/PBOA is anticipated because oxidation of VOC precursors from BB and PB can surely contribute to the observed OOA too. Did BBOA and PBOA decrease substantially (being converted) in either absolute or relative sense during the period that aging to SOA was likely occurring?

2. The off-line analyses: 1) the authors mentioned in the Methods section that detailed chemical speciation approaches such as LC-MS and GC-MS were used. But I do not see any results from them (except levoglucosan, which was used for correlation with BBOA/PBOA); 2) Quartz filters might not be suitable for analysis of inorganic ions because leakages may cause serious positive artefacts, especially for cations such as

sodium ion. Were anions such as sulfate and nitrate measured by off-line IC as well? Why are they not shown in Table 1?

3. Page 11, line 5-10. Levoglucosan had an even higher correlation with PBOA (R-squared = 0.7) than that with BBOA (R-squared = 0.6). Why levoglucosan cannot be used as a tracer for PBOA? If yes, then it seems that this compound is not that specific as a "tracer" to differentiate BBOA and PBOA. The authors may want to discuss a bit on that.

Minor:

1. Page 2, line 13: "wildfire haze between" to "wildfire haze to be between".

2. Page 2, line 17: delete ", in the models".

3. Page 2, line 23: "," after "Aerodyne Inc."

4. Page 2, line 25: "," after "bulk OA"

5. Page 3, line 24: if CE is composition-dependent, it should not be a single factor of 0.5. Around 0.5?

6. Page 4, line 2: "0.2-0.3%". Is this accurate? 20-30%?

7. Page 4, lines 14 and 17: the "a" does not show.

8. Page 4, line 21: "factors solution" to "factor solution".

9. Page 8, line 21-22: I do not think a high standard deviation can indicate the uncertainties of the thermodynamic model. I assume that the standard deviation is from the deviation of all data (pH) from the average pH value, not from running the thermodynamic model in serval ways and averaging all the values. The high standard deviation only indicates the spreading of the pH values during the campaign. Whether the thermodynamic model is accurate or not cannot be reflected by this standard deviation, although I agree with the authors that without NH3 and water absorbed by organics

might cause some uncertainties by ISOROPPIA II.

10. Page 10, line 22-31: the authors did not observe significant correlation between HOA and CO/EC, and suggested that it might be due to limited number of data points or influence by BB or PB (for CO). What about the correlation with NOx?

11. Page 14, line 6-16: how well does ToF-ACSM resolve ion peaks for H:C and O:C calculation? That is, what is the mass spectral resolution of the ToF-ACSM? It might be useful to mention it in the Methods section. Were the H:C and O:C calculated by Aiken's method or the Ambient-Improved method (Canagaratna et al., ACP, 2015)?

---

## Referee Comment (RC2) · Anonymous Referee #2 · 3 Sep 2018

The manuscript entitled, "Dominant contribution of oxygenated organic aerosol to haze particles from real-time observation in Singapore during an Indonesian wildfire event in 2015" by Budisulistiorini et al presents a comprehensive set of measurements on the organic and inorganic chemical composition of particulate matter in Singapore influenced by Indonesian wildfire, using both online and offline techniques. The results highlight a large fraction ($\sim$50% of total OA) of oxygenated OA (OOA) during the haze episodes, indicating the importance of POA oxidation and SOA formation for wildfire haze. In general, I found this manuscript is well written, and most findings follow from

the author's analysis. There is just one issue that may preclude publication of current version in ACP.

Major concern:

pH calculations. The authors calculated pH using thermodynamic model ISORROPIA-II with input of particle-phase concentrations of sulfate, nitrate, chloride, and ammonium. They show that the particles are highly acidic as indicated by an average pH of 1.2. This is unexpectedly low as peat burning also co-emit ammonia. However, as already noted by the authors, the unavailability of gas-phase ammonia data may largely bias the calculated pH. I also agree with Reviewer #1 that this bias is not reflected in the standard deviation. In my point of view, these pH calculations without constraints from gas-phase measurements do not add much value to this manuscript, and therefore could be removed from the manuscript. The calculated LWC, however, might be still useful as it is not sensitive to the gas-phase input.

Technical corrections:

Page 4 Line 2: (0.2-0.3%): should this be 20-30 %?

Page 4 Line 14 and Line 17: symbols are not displayed correctly
* * *

---

## Author Comment (AC1) · 15 Oct 2018

We acknowledge the editor and reviewers for providing insightful comments on the manuscript. We have addressed all of the concerns raised by Referee #1 in the following. Original comments from the reviewer are shown in black, our responses are in blue, and updates on the manuscript are shown in italic blue.

Anonymous Referee #1

Review of "Dominant contribution of oxygenated organic aerosol to haze particles from real-time observation in Singapore during an Indonesian wildfire event in 2015" by Budisulistiorini et al.

The authors presented measurement results of particulate matter (PM) in Singapore affected by an Indonesian wildfire event in 2015. A suite of on-line and off-line measurement techniques were used to quantify chemical composition of PM. The major data from on-line ToF-ACSM measurements showed that oxygenated organic aerosol (OOA) contributed ~50% to total OA in non-refractory PM1 (NR-PM1), while biomass burning and peat burning OA (BBOA and PBOA) contributed 30%, with the remaining 20% being hydrocarbon-like OA (HOA) or residual in PMF/ME-2 analysis. By combining on-line and off-line data, the authors concluded that both oxidation of gaseous precursors and aging of primary OA (POA) contributed to secondary OA (SOA) formation in the wildfire episode. For the POA aging, I incline to agree with the authors that it might occur, but I did not see direct evidence from the data (see Major comment #1 below). I would suggest the authors to either provide more evidence to support this statement or tune it down as a mere speculation as I see it. The measurements and data analyses are rigorous and the manuscript is quite well written. The content is surely within the scope of ACP and is to the interest of readers of ACP. I therefore suggest minor revision with a few comments as below.

We thank the reviewer for acknowledging the worth of the study.

Major:

R1C1.

The assertion that SOA was contributed by POA (BBOA and PBOA) aging was only supported by an observation that the OOA factor has slightly higher f60 than normal background levels (page 14, line 3-5). While the OOA factor was surely influenced by BB or PB, whether the influence was merely factor mixing by PMF/ME-2 or truly due to aging processes is not clear. Set aside the fact that statistical analysis like PMF/ME-2 might suffer from factor mixing problems, correlation between OOA and BBOA/PBOA is anticipated because oxidation of VOC precursors from BB and PB can surely contribute to the observed OOA too. Did BBOA and PBOA decrease substantially (being converted) in either absolute or relative sense during the period that aging to SOA was likely occurring?

Authors' response

We thank the reviewer for the insightful comment. We did not mean to emphasize the contribution of POA aging to the OOA factor, which is generally associated with SOA. Both POA aging and SOA formation, as parts of the atmospheric oxidation process, could be the source of the OOA factor. The slightly higher f60 in the OOA mass spectra compared to that of urban background level was an indicator of the possible influence of biomass and/or peatland fires. Unfortunately, in the present study, we were not able to distinguish them. We revised the main text to clarify the discussion.

Pg 12 Lns 5-14:

*"The OOA factor is more pronounced during the P1 period than during the P2 period, suggesting that SOA formation from oxidation of volatile organic compounds (VOCs) emitted from the peatland fire could contribute to the OOA factor. In fact, some SOA precursors such as benzene, toluene, phenol, and isoprene (Claeys et al., 2004; Yu et al., 2014) were identified from field measurements in Kalimantan, Indonesia during the 2015 peatland fire (Stockwell et al., 2016). Water-soluble organic gases (WSOGs), such as glyoxal, methylglyoxal, and isoprene epoxidiol (IEPOX)), produced from oxidation of VOCs could partition to aerosol liquid water, cloud droplets, or fog water and form SOA (Blando and Turpin, 2000; Ervens et al., 2011; Sareen et al., 2017). Additionally, the OOA factor might also be influenced by oxidation of POA (BBOA and PBOA) during P1 period, as indicated by the presence of levoglucosan ion tracer (m/z 60) in the mass spectra of OOA that is slightly higher than that of the typical background (Cubison et al., 2011)."*

Pg 14 Lns 9-11:

*"The changes in f44 versus f60 are possibly due to the atmospheric oxidation processes, such as levoglucosan decay (Cubison et al., 2011; Hoffmann et al., 2010) and SOA formation processes that enhance ion signals at m/z 44."*

Pg 14 Lns 16-23:

*"The $\overline{OS_C}$ of OOA (0.21) shown in Figure 8 is similar to that of low-volatility OOA (LV-OOA, 0.25) described in a previous study (Canagaratna et al., 2015), suggesting that OOA was highly oxidized. The $\overline{OS_C}$ of OOA also closely agreed with that of laboratory-generated toluene-derived SOA (0.10) estimated by Canagaratna et al. (2015). The close agreement between OOA and laboratory studies indicates that SOA formation could potentially contribute to the OOA factor, considering that SOA precursors—for instance, toluene, benzene, xylene, and other VOCs—were emitted at notable levels (0.1–1 g kg-1) from wildfires in Kalimantan, Indonesia (Stockwell et al., 2016). Linear regression analyses of H:C versus O:C for the OA factors resulted in the slope of –0.06. The slope of the OA factors is close to 0, suggesting that POA aging through hydroxyl group oligomerization in dark chemistry (Gilardoni et al., 2016), and/or the addition of highly oxidized carboxylic and/or peroxide groups (Kumar et al., 2016) might also contribute to the OOA factor."*

Regarding conversion of BBOA and PBOA to SOA, we did not observe the process directly because the study location is located downwind of the peatland fire. The POA and VOCs emitted from the peatland fire would have been oxidized during transport (approximately 1 day) when they arrived in the study location. The strong correlations between the time trends of OOA versus BBOA and/or PBOA ($R^2$ ~0.8, Table S4) indicate that the factors are likely formed at a relatively similar time.

R1C2.

The off-line analyses: 1) the authors mentioned in the Methods section that detailed chemical speciation approaches such as LC-MS and GC-MS were used. But I do not see any results from them (except levoglucosan, which was used for correlation with BBOA/PBOA); 2) Quartz filters might not be suitable for analysis of inorganic ions because leakages may cause serious positive artefacts, especially for cations such as sodium ion. Were anions such as sulfate and nitrate measured by off-line IC as well? Why are they not shown in Table 1?

Authors' response

We thank the reviewer for the detail inquiries.

(1) We revised the main text to highlight the use of molecular analysis by LC-MS and GC-MS for identification of OA factors.

Pg 10 Lns 24-28:
*"To support the identification of the OA factors, we compare the OA factors with chemical species identified by offline analyses (i.e., LC-MS, GC-MS, Sunset OC/EC, and IC-MS). Using the offline analyses, we characterized ambient particles tracers, such as levoglucosan and BrC constituents (Budisulistiorini et al., 2017a), EC and inorganic cations. Table S4 presents the correlation between the OA factors and the ambient particles tracers."*

(2) We corrected the measured inorganic species for blank interference. We did not correct them for positive artifacts from the back filter such as what we did for the organic species. However, the positive artifact would not affect the correlation between the inorganic cations with OA factor solution since the positive artifacts less likely affected the time trend of inorganic ions. We have revised the manuscript by adding this detail as follows.

Pg 5 Lns 22-25:

*"The measured OC, EC, WSOC, and inorganic ions are reported in Table 1. The inorganic ions were corrected for blank interference but not corrected for positive artifacts. We use the measured inorganic to identify the source of OA factors from ME-2 analysis. Hence, we mainly compare the time trend of ions with the OA factors, and do not compare their absolute masses"*

We measured sulfate and nitrate from the integrated samples. We did not report them in the manuscript because we have sulfate and nitrate measurements from ToF-ACSM in a higher time resolution (3 min).

Pg 5 Lns 25-28:
*"Additionally, we only reported inorganic cations that are associated with particles emitted from peat and biomass burning, i.e., $Na^+$, $K^+$, $Mg^+$, and $Ca^{2+}$ (Iinuma et al., 2007). For other inorganic ions, such as sulfate and nitrate, we reported those measured by ToF-ACSM because they have a higher time resolution (~ 3 min) than those measured in the integrated samples (sampling time ~23 hour)."*

R1C3.

Page 11, line 5-10. Levoglucosan had an even higher correlation with PBOA (Rsquared = 0.7) than that with BBOA (R-squared = 0.6). Why levoglucosan cannot be used as a tracer for PBOA? If yes, then it seems that this compound is not that specific as a "tracer" to differentiate BBOA and PBOA. The authors may want to discuss a bit on that.

Authors' response

We thank the reviewer for the suggestion. We agree that levoglucosan is also correlated with PBOA. We did not mean that levoglucosan couldn't be used as a tracer for PBOA. Peat itself is a type of soil made from partially decayed vegetations, including woods. Thus, the OA from peat burning also contains levoglucosan. We have revised the text as follows.

Pg 11 Lns 19-22:

*"We found a good correlation ($R^2 =0.6 – 0.7$, Table S4 and Fig. 4b) between PBOA and levoglucosan as well as the sum of BrC constituents characterized from laboratory peat burning aerosol and the ambient samples of the 2015 haze event (Budisulistiorini et al., 2017a)."*

Minor:

R1C4.

Page 2, line 13: "wildfire haze between" to "wildfire haze to be between".

Authors' response

We have fixed the phrase as suggested.

Pg 2 Lns 11-13:

*"By using atmospheric chemistry transport models, previous studies calculated the excess mortality rate associated with the 2015 wildfire haze to be between 11,000 to 100,000 individuals across Indonesia, Malaysia, and Singapore (Crippa et al., 2016; Koplitz et al., 2016). "*

R1C5.

Page 2, line 17: delete ", in the models".

Authors' response

We have deleted the phrase.

Pg 2 Lns 16-17:

*"This lack of study inhibits a detailed treatment of atmospheric chemical processes in the models, including aerosol aging and secondary aerosol formation."*

R1C6.

Page 2, line 23: "," after "Aerodyne Inc."

Authors' response

We have fixed the phrase as suggested.

Pg 2 Lns 22-24:

*"Online aerosol measurement techniques, such as the Aerosol Mass Spectrometer (AMS) and the Aerosol Chemical Speciation Monitor (ACSM) developed by Aerodyne Inc., are useful for investigating atmospheric processing of aerosol particles, due to their high time resolution for chemical characterization of bulk aerosol composition (Jayne et al., 2000; Ng et al., 2011a)."*

R1C7.

Page 2, line 25: "," after "bulk OA"

Authors' response

We have fixed the phrase as suggested.

Pg 2 Lns 25-27:

*"These techniques quantify the chemical characteristics of bulk OA, allowing further multivariate factor analysis of the mass spectra (MS) matrix by positive matrix factorization (PMF) (Ulbrich et al., 2009; Zhang et al., 2011b) or multilinear engine (ME-2) solvers (Canonaco et al., 2013; Crippa et al., 2014)."*

R1C8.

Page 3, line 24: if CE is composition-dependent, it should not be a single factor of 0.5. Around 0.5?

Authors' response

We thank the reviewer for the comment. The CE is a not single-round number. We have fixed the sentence as follows.

Pg 3 Ln 23-26:

*"A composition-dependent collection efficiency (Middlebrook et al., 2012) factor of approximately 0.5 (0.49 ± 0.17) was applied and confirmed by plotting total NR-PM1 mass concentration measured by the ToF-ACSM against PM2.5 mass concentration published on the website of the National Environment Agency (NEA) of Singapore (Fig. S1)."*

R1C9.

Page 4, line 2: "0.2-0.3%". Is this accurate? 20-30%?

Authors' response

We thank the reviewer for the correction. We have fixed the sentence as follows.

Pg 4 Lns 1-2:

*"Hence, the mass difference found in this study (20–30%) could not solely be attributed to particles with an aerodynamic diameter between 1 and 2.5 $\mu$m that were not measured in this study."*

R1C10.

Page 4, lines 14 and 17: the "a" does not show.

Authors' response

We have fixed the typos as follow.

Pg 4 Ln 14: *"The strength of the constraints was adjusted by varying the degree of freedom (a-value)."*

R1C11.

Page 4, line 21: "factors solution" to "factor solution".

Authors' response

We have fixed the phrase as follows.

Pg 4 Lns 21:

*"The optimum factor solution was also examined by their correlations with chemical tracers and reference mass spectra."*

R1C12.

Page 8, line 21-22: I do not think a high standard deviation can indicate the uncertainties of the thermodynamic model. I assume that the standard deviation is from the deviation of all data (pH) from the average pH value, not from running the thermodynamic model in serval ways and averaging all the values. The high standard deviation only indicates the spreading of the pH values during the campaign. Whether the thermodynamic model is accurate or not cannot be reflected by this standard deviation, although I agree with the authors that without NH3 and water absorbed by organics might cause some uncertainties by ISOROPPIA II.

Authors' response

We thank the reviewer for the discussion. It is true that the standard deviation was calculated from the deviation of all data points from the average. Thus, it could not be associated with the uncertainties of the thermodynamic model. We also acknowledged the lacking of NH3 and organic water would have caused uncertainties in the hydronium ion ($H^+_{activity}$). Therefore, we decided to omit the pH estimation from the manuscript and only reported the aerosol liquid water content from the ISORROPIA estimation. The sentences were revised as follows.

Pg 8 Ln 24-27:

*"Using a thermodynamic model (ISORROPIA-II, Fountoukis and Nenes, 2007), we estimated aerosol liquid water content (LWC) to be $38.1 \pm 5.6$ and $41.9 \pm 5.6$ mol L-1, during P1 and P2 periods respectively (Table S9). Aerosol acidity estimation was omitted because we could of the lacking of NH3 and organic water in the model's input data (Budisulistiorini et al., 2017b; Weber et al., 2016)."*

R1C13.

Page 10, line 22-31: the authors did not observe significant correlation between HOA and CO/EC, and suggested that it might be due to limited number of data points or influence by BB or PB (for CO). What about the correlation with NOx?

Authors' response

We thank the reviewer for the discussion. We, unfortunately, did not measure NOx. The National Environment Agency published $NO_2$ data but not NOx data. We found no correlation between the HOA and $NO_2$ data ($R^2 = 0.02$).

R1C14.

Page 14, line 6-16: how well does ToF-ACSM resolve ion peaks for H:C and O:C calculation? That is, what is the mass spectral resolution of the ToF-ACSM? It might be useful to mention it in the Methods section. Were the H:C and O:C calculated by Aiken's method or the Ambient-Improved method (Canagaratna et al., ACP, 2015)?

Authors' response

We thank the reviewer for the discussion. ToF-ACSM has mass-to-charge resolution $M/\Delta M = 500$ and can measure up to $m/z$ 300 (Fröhlich et al., 2015). The H:C and O:C were calculated following the Improved-Ambient method for unit mass resolution AMS by Canagaratna et al. (2015). We have clarified the point in the revised manuscript.

Pg 3 Lns 18-21:

*"The aerodynamic lens mounted in the ToF-ACSM allowed only PM1 to penetrate into the vacuum chamber (Fröhlich et al., 2013, 2015). The ToF-ACSM has mass-to-charge resolution $M/\Delta M = 500$ and can measure up to m/z 300. Data were obtained with a time resolution of 200 s using IgorDAQ v2.0.20 and the data were analyzed by Tofware version 2.5.6 written in Igor Pro (Wave Metrics Inc., Lake Oswego, Oregon)."*

Pg 14 Lns 10-12:

*"We estimated the ratios of H:C and O:C the Improved-Ambient method for unit mass resolution suggested by Canagaratna et al. (2015) and average $\overline{OS_C}$ following Kroll et al. (2011). The estimated H:C versus O:C as well as $\overline{OS_C}$ of the bulk OA measurements and the OA factors were plotted on a van Krevelen (VK) diagram (Heald et al., 2010) in Figure 8."*

---

## Author Comment (AC2) · 15 Oct 2018

We acknowledge the editor and reviewers for providing insightful comments on the manuscript. We have addressed all of the concerns raised by the Referee #2 in the following. Original comments from the reviewer are shown in black, our responses are in blue, and updates on the manuscript are shown in italic blue.

Anonymous Referee #2

The manuscript entitled, "Dominant contribution of oxygenated organic aerosol to haze particles from real-time observation in Singapore during an Indonesian wildfire event in 2015" by Budisulistiorini et al presents a comprehensive set of measurements on the organic and inorganic chemical composition of particulate matter in Singapore influenced by Indonesian wildfire, using both online and offline techniques. The results highlight a large fraction (50% of total OA) of oxygenated OA (OOA) during the haze episodes, indicating the importance of POA oxidation and SOA formation for wildfire haze. In general, I found this manuscript is well written, and most findings follow from the author's analysis. There is just one issue that may preclude publication of current version in ACP.

We thank the reviewer for acknowledging the worth of the study.

Major concern:

R2C1

pH calculations. The authors calculated pH using thermodynamic model ISORROPIAII with input of particle-phase concentrations of sulfate, nitrate, chloride, and ammonium. They show that the particles are highly acidic as indicated by an average pH of 1.2. This is unexpectedly low as peat burning also co-emit ammonia. However, as already noted by the authors, the unavailability of gas-phase ammonia data may largely bias the calculated pH. I also agree with Reviewer #1 that this bias is not reflected in the standard deviation. In my point of view, these pH calculations without constraints from gas-phase measurements do not add much value to this manuscript, and therefore could be removed from the manuscript. The calculated LWC, however, might be still useful as it is not sensitive to the gas-phase input.

Authors' response

We thank the reviewer for the insightful suggestion. We acknowledged that the pH estimation is lacking in several ways, especially since we were not able to constraint the gas-phase measurements. We have omitted the pH estimation from the manuscript and instead added the estimation of LWC from ISORROPIA into Table S9. The sentences have been revised as follows.

Pg 8 Lns 24-27:

*"Using a thermodynamic model (ISORROPIA-II, Fountoukis and Nenes, 2007), we estimated aerosol liquid water content (LWC) to be 38.1 ± 5.6 and 41.9 ± 5.6 mol L-1, during P1 and P2 periods respectively (Table S9). Aerosol acidity estimation was omitted because we could of the lacking of NH3 and organic water in the model's input data (Budisulistiorini et al., 2017b; Weber et al., 2016)."*

Technical corrections:

R2C2

Page 4 Line 2: (0.2-0.3%): should this be 20-30 %?

Authors' response

We thank the reviewer for the correction. We have revised the typos as shown in our answer for R1C9.

R2C3

Page 4 Line 14 and Line 17: symbols are not displayed correctly

Authors' response

We thank the reviewer for the correction. We have revised the typos as shown in our answer for R1C10.